# Obstructive airway disorders affecting individuals with hereditary hemorrhagic telangiectasia: A database review

**Tristan Sinnatamby[1], Jennifer LaBranche[1,2], Maxine Farr-Jones[1,3], Tiana Fenske[2], Dilini Vethanayagam** [1,3]*

**1** Department of Medicine (Pulmonary), University of Alberta, **2** Faculty of Science, University of Alberta, **3** Edmonton HHT Center, Clinical Sciences Building, University of Alberta, Edmonton, Alberta, Canada

* dilini@ualberta.ca

## Abstract

### Rationale and objective

Dyspnea is a common symptom with varied causes. Pulmonary vascular involvement in Hereditary Hemorrhagic Telangiectasia (HHT) includes pulmonary arteriovenous malformations (PAVMs) and pulmonary hypertension (PH), both which can result in dyspnea. Additionally, dyspnea is a common symptom in individuals with obstructive airway disorders (OADs) such as asthma, bronchiectasis and chronic obstructive pulmonary disease (COPD). These respiratory conditions are not mutually exclusive, individuals with HHT can have pulmonary vascular involvement and a concomitant OAD. However, the likelihood of this co-occurrence is not currently known. We aimed to determine the prevalence of co-occurrence to improve patient diagnosis and management.

### Methods

We conducted a cross-sectional review of individuals seen in the Edmonton HHT Center as of July 2023 with a definite diagnosis of HHT to assess the proportion of patients with a concomitant OAD (asthma, bronchiectasis, or COPD).

### Results

132 patient charts were included. 55.3% had at least one identified PAVM and 28.0% had a documented OAD (asthma = 15.9%, COPD = 10.6%, Bronchiectasis = 2.3%). More importantly, 18.9% of individuals had both an OAD and a PAVM.

### Conclusions

Dyspnea as a symptom of OADs requires specialized assessment and management. PAVMs in HHT also require specialized care, but involve different treatment

**Data availability statement:** There are ethical and legal issues which prevent the public sharing of minimal data for this study, because the data contain potentially identifiable patient information. Data for this study are available upon request from the University of Alberta's Research Ethics Office via email to the corresponding author (dilini@ualberta.ca) or through their website (https://www.ualberta.ca/en/research/services/research-ethics/index.html) for researchers who meet the criteria for access to confidential data.

**Funding:** The author(s) received no specific funding for this work.

**Competing interests:** The authors have declared that no competing interests exist.

approaches. It is important to identify individuals who have both HHT and an OAD to improve management which takes both OADs and pulmonary vascular disorders into account when assessing HHT patients with dyspnea. Early recognition creates better precision health and more effective care of individuals with HHT and dyspnea.

## Introduction

Dyspnea is a common symptom which can significantly impact an individual's quality of life [1]. This can result in patients presenting to different health care environments for assessment including hospitals, emergency departments and primary care settings [1,2]. Dyspnea is a broad term used to describe breathing difficulties ranging from effortful breathing to a sensation of shortness of breath or air hunger [1,3]. There are many possible causes of dyspnea, ranging from pneumonia or pulmonary embolism resulting in acute dyspnea, to chronic dyspnea from chronic obstructive pulmonary disease (COPD) or pulmonary right-to-left shunts, as seen in Hereditary Hemorrhagic Telangiectasia (HHT) [3]. Dyspnea has been reported as a symptom in as many as one third of HHT patients, and has been reported with an even higher prevalence amongst HHT patients with pulmonary arteriovenous malformations (PAVMs) [4].

### HHT

HHT is an autosomal dominant vascular disorder occurring in approximately 1 in 5 000 people in North America, with a higher prevalence of 1 in 3 800 people in Alberta [5–7]. HHT has variable expressivity between individuals with the potential to involve multiple different organ systems [5]. Recurrent epistaxis is the most common symptom of HHT, with epistaxis and telangiectasia in the nasal mucosa reported in over 90% of individuals with HHT [8]. Mucocutaneous telangiectasiae are vessels that are at an increased risk of bleeding and/or rupturing – causing epistaxis when present in the nasal mucosa and causing gastro-intestinal (GI) bleeding when present in the GI mucosa (seen in 15% of patients) [9,10]. Telangiectasiae may also be present on the skin or buccal mucosa, which is the case in approximately 75% of individuals with HHT [10]. Other possible manifestations of HHT involve clinically significant arteriovenous malformations (AVMs), which are direct arteriovenous connections that bypass the capillary bed [7,9]. The vessels in these locations are under high pressure and may rupture leading to significant complications such as strokes [6,9,10]. AVMs can occur in various organ systems including hepatic AVMs (30–70% of patients), pulmonary AVMs (35–40 of patients), cerebral AVMs (10–20% of patients) and spinal AVMs (<1% of patients) [6,7]. Proper organ screening for AVMs after a diagnosis of HHT is critical to effective patient management and prevention of complications [7].

The clinical diagnosis of HHT ("Curaçao Criteria") requires three or four of the following to be present: a) first-degree relative with HHT, b) presence of mucocutaneous telangiectasia, c) evidence of visceral AVMs and d) recurrent epistaxis [5,7]. If only two criteria are present, this reflects a clinical diagnosis of "possible HHT" [11].

Alternatively, a diagnosis can be established with the presence of a pathogenic mutation found through molecular genetics testing in one of the identified HHT genes [11]. Molecular genetic testing for HHT diagnosis is beneficial for asymptomatic individuals with a known family history of HHT and a known pathogenic mutation.[7] As outlined in Online Mendelian Inheritance in Man (OMIM), HHT currently has 6 different genes associated with mutations resulting in the disease [12]. These genes include most commonly, the endoglin gene (*ENG*; chr. 9q34 "HHT1"), Activin receptor-like kinase 1 gene (*ACVRL1*); chr. 12q13 "HHT 2") and the SMAD family member 4 gene (*SMAD4*; chr. 18q24 "JP/HHT"); and less commonly the *GDF2 gene* on chr. 10q11 [HHT5]) [7,12]. There are differences between HHT phenotypes in the frequency and age of onset of various HHT complications such as PAVMs and epistaxis [7,13].

**PAVMs in HHT.** PAVMs can cause numerous complications if they remain untreated. Hypoxemia can result from right-to-left shunting of blood through PAVMs as it bypasses pulmonary capillaries, thereby impairing gas exchange [10]. The blood that is shunted through these PAVMs also bypass the normal filtration process provided by capillary beds and can result in paradoxical emboli that have the potential to cause strokes or cerebral abscesses in the case of septic emboli [10,14]. The PAVMs themselves are also at risk of hemorrhage into the surrounding lung tissue and pleural cavity [10]. Additionally, dyspnea, hemoptysis, digital clubbing and chest pain may all be experienced by patients as a result of PAVMs [10]. The preferred method of treating PAVMs is with embolization, which reduces right-to-left shunting and in turn reduces the associated symptoms and complications [14].

**Pulmonary Hypertension in HHT.** In addition to the direct impact on lung function through the development of PAVMs, HHT can also impact lung function through the development of pulmonary hypertension (PH). PH is defined by a mean pulmonary arterial pressure of 25 mmHg or more, and is classified into 5 groups [15]. There are two different mechanisms for the etiology of PH in HHT patients, both which are a part of group 1, pulmonary arterial hypertension (PAH) [15]. The most common mechanism in HHT populations is pulmonary hypertension due to hepatic AVMs resulting in increased cardiac output and increased blood flow through the pulmonary circulation leading to elevated pressure in the pulmonary vasculature [16,17]. Less commonly, HHT can cause PH due to remodeled pulmonary vascular beds leading to increased resistance, low cardiac output and subsequent right heart failure [16,17]. Various studies have found that PH suggested by echocardiography-occurs in 4–20% of HHT patients [18]. A study by Sopeña et al [19] found that HHT patients who have PH die at a significantly younger age than those without, making PH an important predictor of both morbidity and mortality to identify in this population.

HHT can cause respiratory symptoms due to PAVMs and/or PH. However, the presence of pulmonary vascular disease does not preclude the co-existence of other forms of lung disease which may further contribute to the disease burden. Obstructive airway disorders (OADs) (i.e., asthma, COPD and bronchiectasis) are a common cause of dyspnea in the general population and can result in significant morbidity and mortality [20] OADs may co-exist with vascular lung disease. Additionally, symptoms such as dyspnea can often be misdiagnosed as an OAD if a detailed symptom history is not obtained; this includes disorders such as pulmonary vascular disease from HHT [20] The prevalence of OADs in the general population and their impact on lung function will be discussed below.

**Asthma.** Asthma is a common chronic inflammatory disorder of the airways, affecting 8.7% of Canadians over the age of 12 in 2020, according to Statistics Canada [21]. It is characterized by symptoms of dyspnea, coughing, wheezing and chest tightness which result from airflow limitation and airway hyper-responsiveness [22,23]. Triggers that worsen asthma can be allergenic (i.e., animals, outdoor pollens, indoor/outdoor molds) as well as non-allergenic (i.e., exercise, cold air, humidity) [22]. Asthma is a complex multifactorial, polygenic disorder, meaning that numerous genes and environmental factors play a role in the development of asthma [24]. Many candidate genes have been proposed as susceptibility genes for asthma including Filaggrin, Prostaglandin D2 Receptor gene and Interleukin 1 Receptor-Associated Kinase 3 gene – among more than 100 other confirmed genes and numerous suspected genes [23].

**Chronic Obstructive Pulmonary Disease (COPD).** COPD is another chronic inflammatory disorder of the airways, occurring in approximately 12.1% of Canadians aged 35 and older in 2020, according to Statistics Canada [25]. COPD

presents with similar symptoms as asthma (i.e., dyspnea, cough, sputum production), and is characterized by chronic bronchitis, destruction of small airways and enlarged alveoli [2,26]. Although a major cause of COPD is cigarette smoking, it can also be caused by alpha-1 antitrypsin deficiency, exposure to house-hold fumes and occupational exposures [2].

**Bronchiectasis.** Bronchiectasis is yet another chronic inflammatory disorder of the airways also characterized by similar symptoms as asthma (i.e., dyspnea, cough, sputum production) as well as airway wall architectural distortion and dilatation [27].

As outlined above, dyspnea can occur from pulmonary vascular disorders such as PAVMs and pulmonary hypertension (PH) as well as from more common conditions such as OADs, each with different management strategies. Therefore, differentiating the cause of dyspnea within an individual is important. The frequency of having an OAD along with a diagnosis of HHT, particularly in HHT patients with underlying pulmonary vascular manifestations, is unknown. The co-presence of OADs, which are very common, along with HHT and associated pulmonary vascular diseases, can impact management of these patients and complicate care over time if not recognized. The objective of this study is to establish the prevalence of asthma, COPD and bronchiectasis within the Edmonton HHT population to determine if there is a need for specific screening and treatment protocols for OADs amongst HHT populations.

## Methods

This study utilized the Edmonton HHT Database (single site), which captures patients seen at the Edmonton HHT Center. The data were accessed from the HHT registry and collated between July 18–30, 2023. Accessed data were anonymized.

The HHT database included 525 patients, however only adult and pediatric patients who had provided consent and had confirmed HHT based on clinical criteria or molecular genetics diagnosis were included in the study, providing a total of 132 patients. Any individuals for whom consent could not be obtained or who did not have confirmed HHT were excluded from the study.

Patient demographics and information related to HHT respiratory manifestations and OADs were analyzed. The presence of an OAD was assessed based on respiratory physician clinical diagnosis confirmed with pulmonary physiology testing.

This study was approved by the University of Alberta, Health Research Ethics Board – Health Panel, as an amendment to the existing Edmonton HHT Database ethics.

## Results

As seen in Table 1, A total of 132 subjects with HHT were included in the database review. Of this, 55.3% had at least one identified PAVM. The prevalence of OADs in this HHT population was 28.0%. Impressively, almost one fifth (18.9%) of subjects had both an OAD and a PAVM. (Fig 1)

Of the 132 HHT subjects captured, 15.9% had a diagnosis of asthma, 10.6% had a diagnosis of COPD and 2.3% had a diagnosis of bronchiectasis (Fig 2). The Canadian estimate (excluding the territories) of asthma prevalence provided by Statistics Canada is 8.7% of Canadians over the age of 12 in 2020 [21]. To compare this with the prevalence of asthma in our sample, we calculated the prevalence of asthma among HHT patients over the age of 12, giving an adjusted prevalence of 16.9%. The estimate provided by Statistics Canada of the prevalence of COPD in Canada is 12.1% of Canadians aged 35 and older in 2020 [25]. To compare this with the prevalence of asthma in our sample, we calculated the prevalence of COPD among HHT patients over the age of 35, giving an adjusted prevalence of 13.33%. Bronchiectasis has an unknown prevalence in the general Canadian population and as such we could not compare our data to the general Canadian population statistics.

The prevalence of PH was also assessed and was found to occur in 7.6% of patients included in the study. Amongst the 10 individuals documented with PH, 50% had an identified PAVM as well.

**Table 1. The prevalence of pulmonary vascular involvement and OADs within subjects with HHT from the Edmonton HHT Center database.**

| HHT Subject Group | | Number | Percentage |
|---|---|---|---|
| **Subjects in HHT Database** | | 525 | |
| **Subjects with Definite HHT** | | 132 | n/a |
| *of those with Definite HHT*: | | | |
| | Male | 55 | 41.7 |
| | Female | 75 | 56.8 |
| **Subjects with dyspnea** | | 39 | 29.5 |
| **Subjects with PH** | | 10 | 7.6 |
| **Subjects with PAVM+PH** | | 5 | 3.8 |
| **Subjects with PAVM** | | 73 | 55.3 |
| **Subjects with no PAVM** | | 53 | 40.2 |
| **Subjects with PAVM+dyspnea** | | 26 | 19.7 |
| **Subjects with OAD** | | 37 | 28 |
| | Asthma | 21 | 15.9 |
| | COPD | 14 | 10.6 |
| | Bronchiectasis | 3 | 2.3 |
| **Subjects with OAD** | | 69 | 52.3 |
| **Subjects with PAVM+OAD** | | 25 | 18.9 |
| **Subjects without PAVM or OAD** | | 23 | 17.4 |

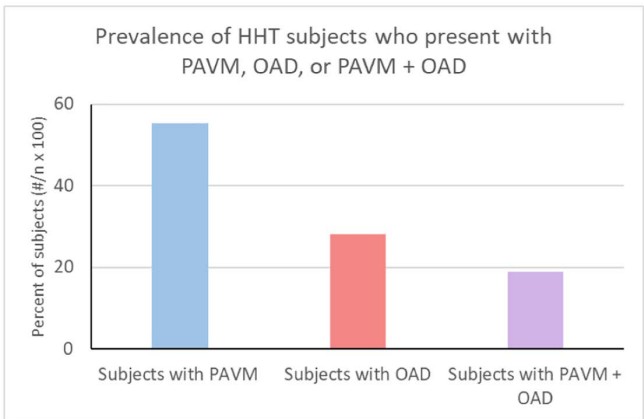

**Fig 1. The prevalence of HHT patients with a PAVM, an OAD, and both a PAVM and an OAD.**

## Discussion

There are many possible organ systems in which HHT can manifest, one of which is the respiratory system – most notably as a PAVM or PH. PAVM's may result in dyspnea, hypoxemia and several other complications. These symptoms can be further complicated by the presence of an OAD such as asthma, COPD or bronchiectasis. It is important to accurately diagnose both HHT as well as any co-occurring airway disorders to effectively manage these patients. Understanding the complexity of a patients' respiratory disease(s) is vital to proper detection and management of their overall disease burden.

In our study we found over one half (55.3%) of the patients had at least one identified PAVM, which is higher than the estimated prevalence of PAVMs in other HHT populations, which is 35–40% [6]. The increased rate of PAVMs in our

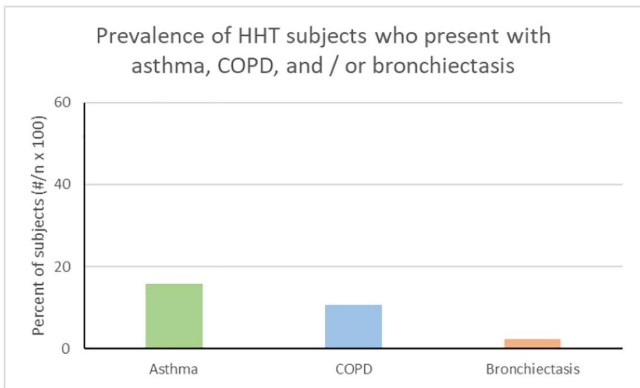

**Fig 2. The prevalence of each specific OAD amongst the HHT patients included in our study.**

population could be due to a number of factors including: small sample size, an increased incidence of PAVMs within our Edmonton HHT Center region or an underestimate of PAVM occurrence across the general Canadian HHT population. Additionally, of the 10 HHT subjects who had PH, half also had at least one PAVM.

A high rate of OADs (asthma, COPD, bronchiectasis) within this population was also noted. There are no good estimates of bronchiectasis prevalence in North America; estimates of asthma and COPD are available for the general Canadian population. The adjusted prevalence of asthma found in our study of 16.9% amongst individuals over the age of 12 was much higher than the Canadian estimate of 8.7% among the same age group.[21] As well, the adjusted prevalence of COPD found in our study of 13.33% amongst individuals over the age of 35 was also higher than the Canadian estimate of 12.1% in the same age group [24]. The Canadian prevalence rates of asthma and COPD provide a comparison point but are only estimates and as such the statistical significance of the difference between our prevalence and that in the overall Canadian population cannot be determined. The observed increased prevalence of asthma and COPD in the HHT population included in our study compared to the general Canadian population may be due to a number of factors. Due to the complicated presentations of patients with both HHT and an OAD as well as the under-recognition of HHT, many patients may be misdiagnosed with an OAD without HHT being recognized as an underlying cause.[20] In this case it is important for patients newly diagnosed with HHT to be re-evaluated for a pre-diagnosed OAD in light of the HHT diagnosis.

One limitation of our study, particularly given the rural/remote nature of the areas served by many of the subjects in our health jurisdiction, is the relatively small number of subjects captured. Further work could benefit from a larger sample size that looks beyond the Edmonton HHT center population. Another limitation is that molecular genetics and their link to the concomitant presence of OAD was not evaluated. Unfortunately this data was not accessible for this study; additionally, the majority of patients included did not have a mutation identified through molecular genetics or in other circumstances not done. Future studies should include genetic analysis.

This project draws attention to an under-recognized group of high-risk respiratory diagnoses that may otherwise be missed. All the pulmonary conditions examined complicate the care and diagnosis of individuals with HHT, especially within the PAVM subgroup, thus it is important to identify these individuals in order to provide appropriate diagnosis and care. This study highlights the need to screen for both HHT and OADs when looking at patient groups at high risk for respiratory diseases. Within the clinical setting, these results highlight the need for HHT centers to establish standard screening protocols for OADs, to better manage and treat HHT patients who have pulmonary manifestations. This should especially be considered when patients present with signs and symptoms that are commonly observed in both HHT pulmonary manifestations and OADs, such as dyspnea and chest tightness. Additionally, screening for HHT in individuals

who have an OAD should be considered in a case-by-case manner for patients with a higher risk of also having this genetic disease (i.e., other organ manifestations of HHT, treatment-refractory OAD). The interconnection between OAD and HHT indicates clinical the importance of comprehensive and specific screening for respiratory diseases and resultant treatment of the different diagnosed issues. Each respiratory disease requires different treatment and management plans making this vital to providing individualized and effective care to patients.

## Conclusions

OADs require specialized care and management. PAVMs in HHT also require specialized care but are treated differently. Correct diagnosis and identification of individuals with HHT who also have underlying OAD will improve overall effective management for both disease processes, hopefully resulting in fewer complications and improved quality of life over time. We need to take OADs as well as pulmonary vascular disorders into account when assessing HHT patients with dyspnea. While there is no current evidence that the conditions are linked, it would be useful to explore a possible common pathogenesis in the future.

## Author contributions

**Conceptualization:** Dilini Vethanayagam.

**Data curation:** Jennifer LaBranche, Tiana Fenske, Dilini Vethanayagam.

**Formal analysis:** Tristan Sinnatamby, Dilini Vethanayagam.

**Investigation:** Jennifer LaBranche.

**Methodology:** Jennifer LaBranche, Dilini Vethanayagam.

**Project administration:** Maxine Farr-Jones.

**Validation:** Tristan Sinnatamby.

**Writing – original draft:** Tristan Sinnatamby.

**Writing – review & editing:** Jennifer LaBranche, Maxine Farr-Jones, Tiana Fenske, Dilini Vethanayagam.

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
