## [Decision Letter · Decision Letter 0]

7 Jan 2025

Dear Dr. Vethanayagam,

Thank you for submitting your manuscript to PLOS ONE. After careful consideration, we feel that it has merit but does not fully meet PLOS ONE’s publication criteria as it currently stands. Therefore, we invite you to submit a revised version of the manuscript that addresses the points raised during the review process.

We look forward to receiving your revised manuscript.

Kind regards,

Mehmet Baysal

Academic Editor

PLOS ONE

2. In the online submission form you indicate that your data is not available for proprietary reasons and have provided a contact point for accessing this data. Please note that your current contact point is a co-author on this manuscript. According to our Data Policy, the contact point must not be an author on the manuscript and must be an institutional contact, ideally not an individual. Please revise your data statement to a non-author institutional point of contact, such as a data access or ethics committee, and send this to us via return email. Please also include contact information for the third party organization, and please include the full citation of where the data can be found.

Reviewers' comments:

Reviewer's Responses to Questions

**Comments to the Author**

1. Is the manuscript technically sound, and do the data support the conclusions?

Reviewer #1: Yes

Reviewer #2: Yes

2. Has the statistical analysis been performed appropriately and rigorously?

Reviewer #1: N/A

Reviewer #2: N/A

3. Have the authors made all data underlying the findings in their manuscript fully available?

Reviewer #1: Yes

Reviewer #2: Yes

4. Is the manuscript presented in an intelligible fashion and written in standard English?

Reviewer #1: Yes

Reviewer #2: Yes

Reviewer #1: This is a short and comprehensive cross-sectional review of individuals seen in Edmonton (Canada) as of July 2023 with a definite diagnosis of hereditary hemorrhagic telangiectasia (HHT), an inherited vascular disease with a relatively high prevalence (1:5,000-8,000 people worldwide, and even higher in Alberta) among rare diseases. HHT patients usually present with pulmonary arteriovenous malformations (PAVMs) and pulmonary hypertension (PH), both of which can result in dyspnea. The goal is to assess in HHT patients the possible overlapping between HHT manifestations and obstructive airway disorders (bronchiectasis, asthma or chronic obstructive pulmonary disease-COPD). The Introduction section properly describes the clinical manifestations involving the respiratory system of the different studied conditions, including HHT, bronchiectasis, asthma or COPD. The authors find that among 132 HHT patients: (i) 55.3% had at least one PAVM; (ii) 28.0% had obstructive airway disorders (asthma=15.9%, COPD= 0.6%, Bronchiectasis =52 2.3%); and (iii) 18.9% had both an obstructive airway disorder and PAVMs. This study suggests that among HHT patients, the correct diagnosis of individuals who also have obstructive airway disorders is key to improve the overall management for both disease processes. It also emphasizes that patients at high risk for respiratory diseases should be screened for HHT. These findings are interesting and relevant to the field of HHT. The manuscript is well written in straightforward fashion. The different sections are presented fluently, while figures and tables provide clear information. Independently of the scientific merit of the manuscript, the selected target journal (PONE) is addressed to a broad audience of readers, while publication in a clinical journal would have had more impact among expert readers. Several clarifications and minor points should be addressed to improve the quality of the manuscript:

1. -Page 5, lines 103,104. “and less commonly a gene on chr. 5q31 [HHT 3], chr. 7p14 [HHT 4]”. Variants HHT3 and HHT4, with candidate loci on chromosomes 5 and 7, respectively have been definitively ruled out based on recent whole genome sequencing studies (Shovlin et al. J Med Genet. 2024; 61:182-185. doi: 10.1136/jmg-2023-109195). Please modify the sentence accordingly.

2. -Page 5, line 100. “6 different genes”. The putative genes involved in HHT3 and HHT4 have never been identified. In addition, based on the above item, “6 different genes” should be “4 different genes”

3. -Over 90% of HHT patients present mutations in either ENG (HHT1) or ACVRL1 (HHT2) genes. Because HHT1 patients are associated with PAVMs more frequently than HHT2 patients, it would be interesting to analyze the individual correlation data (PAVMs / obstructive airway disorders), within each of these two HHT subsets. Unfortunately, the authors do not present the corresponding genetic analysis of the HHT population under study. This limitation should be experimentally addressed, or at least commented.

4. -In M&M, Results and Discussion sections, the population percentages affected by asthma and COPD in Canada are indicated. Are these figures similar to those in US or European populations? Please comment.

Reviewer #2: Dear Author;

A very valuable evaluation, thank you for your contributions to the literature. However, the current data provides a cross-sectional evaluation and only the frequency of PAVM and OAD in the data analysis is mentioned. The result provides a current regional interpretation. It has no clinical reflection and no relationship with genetic mutations in a personalized way. In addition, the introduction part is quite long and detailed. If the results of PAVM and OAD findings in the presence of genetic mutations in the patients are available, they can be added. Again, the clinical findings of the patients in the presence of these findings can be evaluated.

Best regards

**Do you want your identity to be public for this peer review?** For information about this choice, including consent withdrawal, please see our Privacy Policy

Reviewer #1: No

Reviewer #2: **Yes: ** MD. Ufuk Demirci

---

## [Author Response · Author response to Decision Letter 1]

30 Jul 2025

Reviewer 1:

1. Page 5, lines 103,104. “and less commonly a gene on chr. 5q31 [HHT 3], chr. 7p14 [HHT 4]”. Variants HHT3 and HHT4, with candidate loci on chromosomes 5 and 7, respectively have been definitively ruled out based on recent whole genome sequencing studies (Shovlin et al. J Med Genet. 2024; 61:182-185. doi: 10.1136/jmg-2023-109195). Please modify the sentence accordingly.

Thank you. This was done.

2. Page 5, line 100. “6 different genes”. The putative genes involved in HHT3 and HHT4 have never been identified. In addition, based on the above item, “6 different genes” should be “4 different genes”

Thank you. As with the first comment, this was done and the update you noted was reviewed and included in the reference list (ref 12).

3. Over 90% of HHT patients present mutations in either ENG (HHT1) or ACVRL1 (HHT2) genes. Because HHT1 patients are associated with PAVMs more frequently than HHT2 patients, it would be interesting to analyze the individual correlation data (PAVMs / obstructive airway disorders), within each of these two HHT subsets. Unfortunately, the authors do not present the corresponding genetic analysis of the HHT population under study. This limitation should be experimentally addressed, or at least commented.

Unfortunately the molecular genetics information for each patient was not captured for this study. A note on this limitation has been included within the manuscript discussion section (see lines 262-266).

4. In M&M, Results and Discussion sections, the population percentages affected by asthma and COPD in Canada are indicated. Are these figures similar to those in US or European populations? Please comment.

Compared to Canada, the prevalence of asthma was similar/slightly lower in the US (7.7% overall; 8.0% for 18+). European countries varied more, however ranged between 5.1%-8.2%, which is fairly similar to Canadian statistics.

Compared to Canada, the prevalence of COPD was surprisingly much lower in the US (4.6% for 18+) and in Europe (ranged from 0.1-7.8%).

However, given that asthma and COPD are both influenced by environmental factors, we thought it was most pertinent to compare our Canadian study results to rates within Canada.

Reviewer 2:

A very valuable evaluation, thank you for your contributions to the literature. However, the current data provides a cross-sectional evaluation and only the frequency of PAVM and OAD in the data analysis is mentioned. The result provides a current regional interpretation. It has no clinical reflection and no relationship with genetic mutations in a personalized way. In addition, the introduction part is quite long and detailed. If the results of PAVM and OAD findings in the presence of genetic mutations in the patients are available, they can be added. Again, the clinical findings of the patients in the presence of these findings can be evaluated.

Thank you. Regarding the data analysis comment, PH was also mentioned, along with PAVMs and OAD. These three topics were the only ones being assessed based on the goal of the study outlined in the introduction section.

Given the complexity of the disorders mentioned in this manuscript, the introduction was detailed to ensure the PLOS One audience would understand each disorder assessed within the study. Additionally, creating separate sections within the introduction was aimed at improving comprehension and organization.

Unfortunately the molecular genetics information for each patient was not captured for this study. A note on this limitation has been included within the manuscript discussion section.

A reflection on the clinical relevance of this study was present, particularly in the last paragraph of the discussion section (lines 267-284). Edits have been made to make the reflection more clear.

---

## [Decision Letter · Decision Letter 1]

20 Aug 2025

Obstructive Airway Disorders Affecting Individuals with Hereditary Hemorrhagic Telangiectasia: A Database Review

PONE-D-24-39482R1

Dear Dr. Vethanayagam,

We’re pleased to inform you that your manuscript has been judged scientifically suitable for publication and will be formally accepted for publication once it meets all outstanding technical requirements.

Kind regards,

Mehmet Baysal

Academic Editor

PLOS ONE

Additional Editor Comments (optional):

Reviewers' comments:

Reviewer's Responses to Questions

**Comments to the Author**

Reviewer #1: All comments have been addressed

Reviewer #2: All comments have been addressed

2. Is the manuscript technically sound, and do the data support the conclusions?

Reviewer #1: Yes

Reviewer #2: Yes

3. Has the statistical analysis been performed appropriately and rigorously?

Reviewer #1: I Don't Know

Reviewer #2: Yes

4. Have the authors made all data underlying the findings in their manuscript fully available?

Reviewer #1: Yes

Reviewer #2: Yes

5. Is the manuscript presented in an intelligible fashion and written in standard English?

Reviewer #1: Yes

Reviewer #2: Yes

Reviewer #1: (No Response)

Reviewer #2: Dear Author;

This is a comprehensive cross-sectional study aimed at evaluating hereditary hemorrhagic telangiectasia (HHT) with obstructive airway disorders (bronchiectasis, asthma, or chronic obstructive pulmonary disease (COPD). The article is clearly and understandably presented. It will contribute to the literature with a broad patient population. I believe the article has two significant shortcomings. First, it lacks clinical reflection because it is a cross-sectional study. However, I believe that the results of this study, along with the clinical evaluation of patients, may provide new study opportunities for centers. In this sense, it will provide data that can be cited. Furthermore, a significant lacuna in the present study is the absence of genetic findings. You added this limitation to the article and explained the reason. Additionally, you mentioned that it should be evaluated in future studies, also, strengthened the article by adding other details.

Best regards,

**Do you want your identity to be public for this peer review?** For information about this choice, including consent withdrawal, please see our Privacy Policy

Reviewer #1: No

Reviewer #2: **Yes: ** Ufuk Demirci

---

## [Editor Report · Acceptance letter]

PONE-D-24-39482R1

PLOS ONE

Dear Dr. Vethanayagam,

I'm pleased to inform you that your manuscript has been deemed suitable for publication in PLOS ONE. Congratulations! Your manuscript is now being handed over to our production team.

Kind regards,

on behalf of

Dr. Mehmet Baysal

Academic Editor

PLOS ONE